# Nutrients Drive the Structures of Bacterial Communities in Sediments and Surface Waters in the River-Lake System of Poyang Lake

**Ze Ren [1,2], Xiaodong Qu [1,3],* , Wenqi Peng [1,3], Yang Yu [1,3] and Min Zhang [1,3]**

1   State Key Laboratory of Simulation and Regulation of Water Cycle in River Basin, China Institute of Water Resources and Hydropower Research, Beijing 100038, China; renzedyk@gmail.com (Z.R.); pwq@iwhr.com (W.P.); yuyang@iwhr.com (Y.Y.); zhangmin@iwhr.com (M.Z.)
2   Division of Biological Sciences, University of Montana, Missoula, MT 59812, USA
3   Department of Water Environment, China Institute of Water Resources and Hydropower Research, Beijing 100038, China
*   Correspondence: quxiaodong@iwhr.com

**Abstract:** Lake and its inflow rivers compose a highly linked river-lake system, within which sediment and water are also closely connected. However, our understanding of this linked and interactive system remains unclear. In this study, we examined bacterial communities in the sediments and surface waters in Poyang Lake and its five tributaries. Bacterial communities were determined while using high-throughput 16S rRNA gene sequencing. The results showed significant differences of bacterial communities between sediments and surface waters, as well as between Poyang lake and its tributaries, suggesting that the river-lake system of Poyang Lake provides diverse and distinct habitats for bacterial communities, including lake water, lake sediment, river water, and river sediment. These biomes harbor distinct bacterial assemblages. Sediments harbor more diverse bacterial taxa than surface waters, but the bacterial communities in surface waters were more different across this river-lake system than those in sediments. In this eutrophic river-lake ecosystem, nitrogen and phosphorus were important drivers in sediment bacterial communities. Nitrogen, phosphorus, and dissolved organic carbon, as well as their stoichiometric ratios affected bacterial communities in surface waters. Moreover, network analysis revealed that the bacterial communities in surface waters were more vulnerable to various disturbances than in sediments, due to lower alpha diversity, high complexity of network, and a small number of key taxa (module hubs and connectors). Nutrient variables had strong influences on individual operational taxonomic units (OTUs) in the network, especially in bacterial network in surface waters. Different groups of taxa responded differently to nutrients, with some modules being more susceptible to nutrient variations. This study increased our current knowledge of linked river-lake ecosystems and provided valuable understanding for effective management and protection of these ecosystems by revealing bacterial communities in sediments and surface waters in Poyang Lake and its tributaries, as well as their responses to nutrients variation.

**Keywords:** nutrient; 16S rRNA; alpha diversity; community dissimilarity; network

## 1. Introduction

Microorganisms encompass tremendous diversity [1] and exhibit high compositional and functional variability in freshwater environments [2]. Bacterial communities have a broad genetic diversity in the water body in lake ecosystems [2,3]. As a distinct realm of lake ecosystems, sediments also host a tremendous diversity of microorganisms, which play vital roles in maintaining the benthic food web structure, as well as driving major biogeochemical cycles [4,5]. In aquatic ecosystems,

biogeochemical interactions closely connected sediments and the overlying water body. For example, materials are deposited from water columns to sediments, while decomposition in sediments release dissolved substances into the water body [5]. The metabolic activity of the bacterial community in sediments, such as nutrient store and release, methane production, and iron reduction, can drive the biogeochemical cycles and influence water quality [6,7]. In addition, due to the different chemical and physical environment between water and sediment habitats, the processes driving the variation of bacterial communities are different in water and sediment environments [8].

Lakes and their inflow rivers are also distinct but highly linked habitats in watersheds, despite the differences and connections between water column and sediments [9–12]. Lake and river systems have a large contrast in physical and chemical properties [13], leading to different biogeochemical paradigms and further driving the bacterial communities differently in these distinct ecosystems. On the other hands, linkages among aquatic ecosystems are particularly extensive and they provide opportunities in the exchange of materials [14,15]. Rivers receive nutrients and organic matters from the catchment [16–18]. Consequently, lakes are intimately associated with catchment characteristics through materials that are transported by their tributaries [19–22]. The linkages between lake and river ecosystems have been of great research interests, but the relationships between lake and river bacterial communities are not well understood.

Different habitats usually harbor different assemblages of microorganisms [23–25]. Previous research has demonstrated that bacterial communities in lake sediment and the water column are different [26–29] and they are controlled by a variety of factors, such as dispersal, energy and nutrient availability, and anthropogenic influences [30,31]. In a linked stream-lake ecosystem, bacterial communities in lake water and stream biofilm are taxonomically and functionally distinct [29]. However, in a linked river-lake ecosystem, an integral understanding of the bacterial communities in sediments and surface waters in both the lake and its inflow tributaries is still limited. Bacterial communities would undergo changes to keep subsistence in the distinct physicochemical environments and, in turn, could influence the environment of the linked river-lake ecosystem through the intimate linkages between different habitats. Therefore, the study of bacterial community structures in surface waters and sediments of lakes and its tributaries is crucial in providing insight into ecosystem structures and processes, as well as community assembly rules of river-lake systems.

Here, we studied and compared the bacterial communities in lake sediment (LS), river sediment (RS), lake water (LW), and river water (RW) in the river-lake system of Poyang Lake. Bacterial samples of surface waters and sediments from five major tributaries and Poyang Lake itself were collected in this river-lake ecosystem. Bacterial communities were determined using the high-throughput 16S rRNA gene sequencing. Our object is to reveal the different compositions and driving factors of bacterial communities between Poyang Lake and its tributaries in the habitats of sediments and surface waters of this river-lake system. We hypothesized that (1) in this river-lake system, lake sediment, river sediment, lake water, and river water harbor different bacterial communities and (2) these bacterial communities respond strongly, but differently to nutrients.

## 2. Materials and Methods

### 2.1. Study Area

Poyang Lake (PY) is located in the lower reach of Yangtze River in the northern part of Jiangxi Province. It is the largest freshwater lake (during summer at high water level) in China and is one of the two lakes (the other one is Dongting Lake in Hunan Province) that are directly connected to Yangtze River (Figure 1). Poyang Lake has a surface area over 4000 km$^2$ in the summer [32,33]. The average depth is 8.4 m. There are five rivers, Xiushui (XS), Ganjiang (GJ), Fuhe (FH), Xinjiang (XJ), and Raohe (RH) that feed Poyang Lake and one outlet that connects to Yangtze River (Figure 1). The annual runoff of Poyang Lake is 152.5 billion m$^3$, which accounts for 16.3% annual runoff of Yangtze River. Yangtze River and the inflows of the five tributaries, forming the water-carrying and throughput hydrological

characteristics highly restrict the water level of Poyang Lake [34,35]. The watershed area of Poyang Lake is 162,200 km$^2$, which accounts for 9% of the whole area of Yangtze River basin. Within the watershed, the cultivated land area and forest coverage rate are 34.2% and 63.1% in 2012, respectively. The flood season of Poyang Lake generally occurs at the end of March and last to October [32]. In the summer, Poyang Lake usually takes in the flood from Yangtze River to reduce the flood risks of the downstream area. Due to the variation of tributary inflows and the water exchange (out flow and reverse flow) with the Yangtze River, the water level significantly fluctuates with alternating periods of floods and droughts, resulting in large seasonal variation of water surface area [36,37].

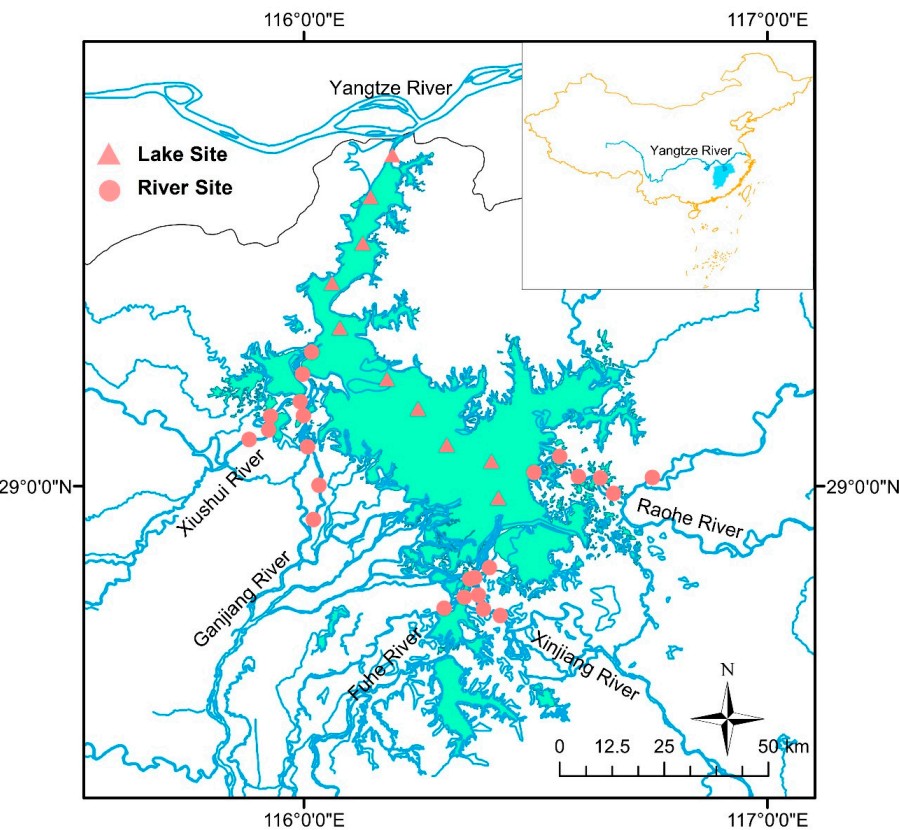

**Figure 1.** Study area and sample sites distribution. Samples (water and sediment) were collected from Poyang Lake (10 sites) and its five major tributaries (24 sites), Xiushui River, Ganjiang River, Fuhe River, Xinjiang River, and Raohe River. The map was created in ArcGIS 14.0 (http://desktop.arcgis.com/en/arcmap/).

## 2.2. Field Sampling

We set up 10 sample sites in Poyang Lake and 24 sample sites in its tributaries (Figure 1). In each sample site, both water (PY.W, FH.W, GJ.W, RH.W, XJ.W, XS.W) and sediment samples (PY.S, FH.S, GJ.S, RH.S, XJ.S, XS.S) were collected for chemical and microbial analyses in early August 2017. The water samples were collected at the depth of 0.5 m using a Van Dorn water sampler. 500 mL water (three replicates for each site) was acid fixed in the field and then transported at 4 °C to the laboratory for chemical analyses. Another 200 mL water was filtered onto a 0.2-µm polycarbonate membrane filter (Whatman, Maidstone, UK) and then immediately frozen in liquid nitrogen in the field and stored in −80 °C freezer in the laboratory until DNA extraction. The sediment samples were collected (at the same site with water sample) using a Ponar Grab sampler. The top 5-cm sediment was collected and homogenized. 45 mL sediment sample (three replicates for each site) was filled in a sterile centrifuge tube and then immediately frozen in liquid nitrogen in the field and stored in −80 °C freezer in the laboratory until DNA extraction. The remaining sediment was refrigerated for chemical analyses.

## 2.3. Chemical Analyses

Total nitrogen (TN) was tested using ion chromatography (EPA 300.0, revision 2.1, 1993) with a prior persulfate oxidation for the water samples [38]. Nitrate ($NO_3^-$) was tested using ion chromatography (EPA 300.0, revision 2.1, 1993). The indophenol colorimetric method analyzed Ammonium ($NH_4^+$) (EPA 350.1, revision 2.0, 1993). The ascorbic acid colorimetric method with a prior oxidation determined total phosphorus (TP) (EPA 365.3, 1978). Ascorbate acid colorimetric determined the soluble reactive phosphorus (RSP) (EPA 365.3, 1978). Dissolved organic carbon (DOC) was analyzed while using a TOC Analyzer (TOC-VCPH, Columbia, MD, USA). Table S1 shows the nutrient concentrations and stoichiometry ratios in river and lake water.

Sediment samples were completely dried in an oven under 60 °C for 3–5 days and homogenized by grinding. The potassium dichromate oxidation spectrophotometric method (HJ615-2011) was used to analyze the total organic carbon (OC). The modified Kjeldahl method determined TN (HJ717-2014). TP was determined by alkali fusion-Mo-Sb Anti spectrophotometric method (HJ632-2011). $NO_3^-$ and $NH_4^+$ were determined using the UV spectrophotometry method with potassium chloride extraction (HJ634-2012). The acid hydrolysis method determined organic nitrogen (ON) [39]. The SMT method determined organic phosphorus (OP) [40]. Table S2 shows the nutrients contents and stoichiometry ratios in river and lake sediment.

## 2.4. DNA Extraction, PCR, and Sequencing

DNA was extracted from the filter and sediment (0.5 g) samples while using the TIANGEN-DP336 soil DNA Kit (TIANGEN-Biotech, Beijing, China) following the manufacturer's protocol. Qubit 2.0 Fluorometer (Invitrogen, Carlsbad, CA, USA) was used to quantify the extracted DNA samples. The high variability regions V3 and V4 regions of 16S rRNA genes were amplified while using the forward primer 347F 5'-CCTACGGRRBGCASCAGKVRVGAAT-3' and the reverse primer 802R 5'-GGACTACNVGGGTWTCTAATCC-3' (GENEWIZ, Inc., South Plainfield, NJ, USA). PCR was performed in a model 2720 thermal cycler (ABI, USA) using the following program: 94 °C initial denaturation for 3 min, 24 cycles of denaturation at 94 °C for 30 s, followed by annealing at 57 °C for 90 s and extension at 72 °C for 10 s, and the final extension at 72 °C for 10 min. Amplified DNA was verified in 1.0 % agarose in 1X TAE buffer and purified using the Gel Extraction Kit (Qiagen, Hilden, Germany). DNA libraries were validated (Agilent 2100 Bioanalyzer, Agilent Technologies, Palo Alto, CA, USA) and then quantified (Qubit 2.0 Fluorometer, Invitrogen, Carlsbad, CA, USA). According to manufacturer's instructions, the DNA libraries were multiplexed and loaded on an Illumina MiSeq instrument (Illumina, San Diego, CA, USA) for sequencing.

## 2.5. Data Analyses

Raw sequence data were processed using the software package QIIME (Quantitative Insights Into Microbial Ecology) 1.9.1 [41]. The forward and reverse reads were merged and then assigned to samples based on barcode. The joined sequences were truncated by cutting off the barcode and primer sequence and they were quality filtered. Sequences that did not fulfill the following criteria were discarded: sequence length < 200 bp, no ambiguous bases, and mean quality score ≥ 20. UCHIME (version 4.2) algorithm [42] was used to compare the sequences with the RDP (Ribosomal Database Project) Gold Database to detect chimeric sequences, which were removed. Subsequently, the effective sequences were grouped into operational taxonomic units (OTUs) against the Greengenes 13.8 database [43] at 97% sequence identity level. The alpha diversity indices were calculated using QIIME, including observed OTUs, Shannon index, and Faith's phylogenetic diversity. Raw sequence data were deposited at the National Center for Biotechnology Information (PRJNA436872, SRP133903).

LMER test (R package lmerTest 3.1) [44] was used to determine the differences of the alpha diversity and relative abundances of dominant phyla (relative abundance > 1%) between the bacterial communities in different habitats. To test the statistical significance of the differences between

bacterial communities in LS, RS, LW, and RW, analysis of variance using distance matrices (ADONIS, 'adonis' function), analysis of similarity (ANOSIM, 'anosim' function), and multi-response permutation procedure analysis (MRPP, 'mrpp' function) were conducted while using R package vegan 2.5-3 [45]. Redundancy analysis (RDA) was used to examine the relationships between bacterial communities and nutrient factors, and the significance of environmental factors was tested using permutation test using R package vegan 2.5-3 [45]. Network analyses were conducted to reveal the co-occurrence patterns of the bacterial communities in sediments and surface waters. OTUs with an average relative abundance > 0.01% and presented in more than half samples were used. The pairwise correlations between OTUs were calculated using the Spearman correlation. The *p*-values were adjusted by FDR correction. Only strong and significant correlations (Spearman's r > 0.8 or r < −0.8, *p* < 0.05) were considered. Network visualization, topological parameters, and modular analysis were made with the R package igraph 1.2.4 [46]. The topological roles (module hubs and connectors) of each of the OTUs were determined according to the within-module connectivity (Zi = 2.5) and the among-module connectivity (Pi = 0.62). Meanwhile, 999 random networks (with the same number of nodes and edges as the real networks) were generated in the igraph package according to the Erdos-Renyi model. Comparisons between two real networks were conducted using the *t*-test and the Z-test was used to conduct comparisons between real networks and their corresponding random networks [47,48]. All of the statistical analyses were carried out in R 3.4.1 [49].

## 3. Results

### 3.1. Alpha Diversity

In this study, we generated 9,952,230 raw sequences for 68 libraries. A total of 2,900,931 reads were retained after quality filtering and 7410 OTUs were detected at a 97% nucleotide sequence identity threshold. In general, LS and RS had a significantly higher alpha diversity than LW and RW (Figure 2), which suggests that, in both Poyang Lake and its tributaries, sediments harbor more diverse microorganisms than the surface waters (LS vs. LW, RS vs. RW). However, the alpha diversity of sediment bacterial communities was not significantly different between rivers and the lake (LS vs. RS), except for higher observed OTUs and phylogenetic diversity in RH than in FH (Figure 2). Moreover, bacterial alpha diversity in surface waters have shown some differences across the river-lake system.

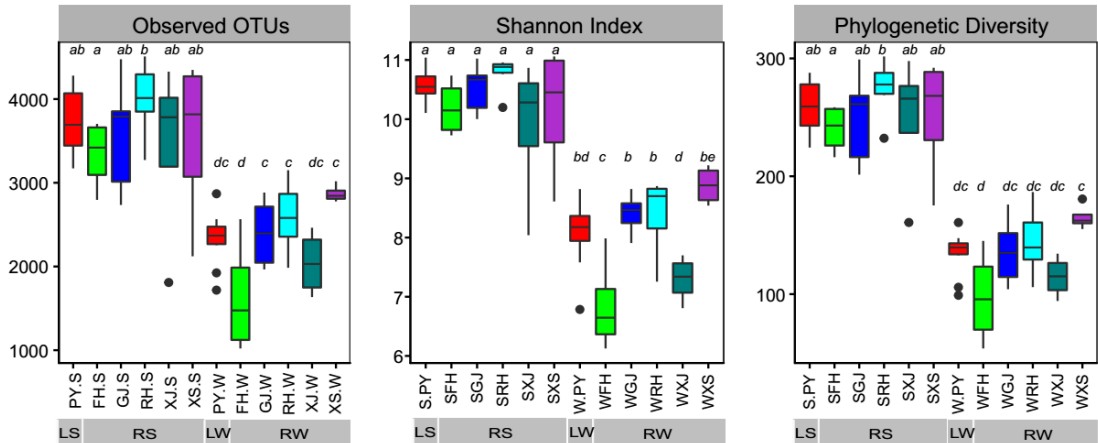

**Figure 2.** Alpha diversity indexes (observed operational taxonomic units (OTUs), Shannon index, and phylogenetic diversity) of bacterial communities in lake sediment (LS), river sediment (RS), lake water (LW), and river water (RW) in the river-lake system of Poyang Lake. Different letters indicate significant differences at *p* < 0.05 level.

*3.2. Community Structure*

Proteobacteria were the dominant bacterial phyla (relative abundance > 1%) in LS and RS, followed by Acidobacteria, Bacteroidetes, Nitrospirae, Chloroflexi, Chlorobi, Gemmatimonadetes, WS3, Cyanobacteria, Actinobacteria, and Verrucomicrobia (Figure 3). However, Proteobacteria, Cyanobacteria, Bacteroidetes, Actinobacteria, and Thermi were the dominant bacterial phyla (relative abundance > 1%) in LW and RW (Figure 3). The most abundant family in LS and RS was Thermodesulfovibrionaceae (Figure S1). The most abundant family in LW and RW were Moraxellaceae (Figure S1). Non-parametric statistical tests further demonstrated the dissimilarities between sediments and surface waters and between Poyang lake and its tributaries (ADONIS, ANOSIM, and MRPP) (Figure 4). When comparing sediments to surface waters, bacterial communities in LS and RS were significantly different to LW and RW (LS vs. LW, RS vs. RW), respectively (Figure 4a), suggesting that sediments and surface waters are different habitats that harbor distinct bacterial communities. Moreover, when comparing Poyang Lake to its tributaries, the bacterial communities were also significantly different between LS and RS (LS vs. RS, Figure 4b), as well as between LW and RW (LW vs. RW, Figure 4c), which suggests that Poyang Lake and its tributaries had different bacterial communities no matter in sediment or in water. However, in comparing different tributaries, bacterial communities in water exhibited higher dissimilarities with each other (Figure 4c) than that of bacterial communities in sediments (Figure 4d), suggesting that bacterial communities in the surface waters were more different among tributaries.

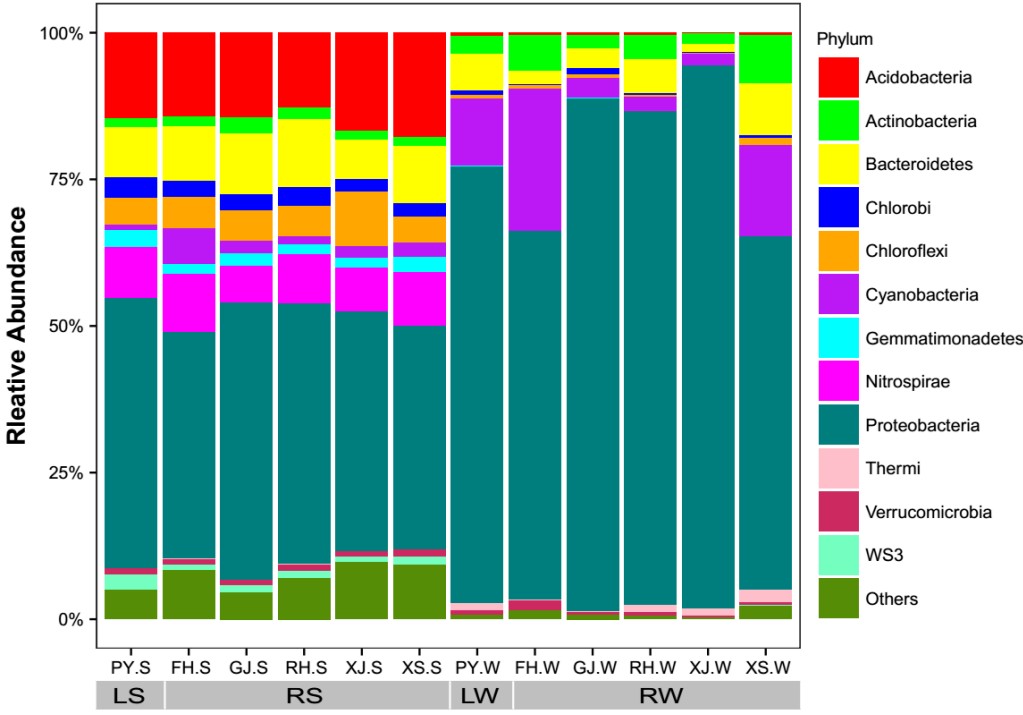

**Figure 3.** Relative abundance of microorganisms at phylum level in lake sediment (LS), river sediment (RS), lake water (LW), and river water (RW) in the river-lake system of Poyang Lake. Only the phyla that had a relative abundance > 1% in either habitat are shown. "Others" represent the phyla with a relative abundance < 1% as well as the unsigned OTUs.

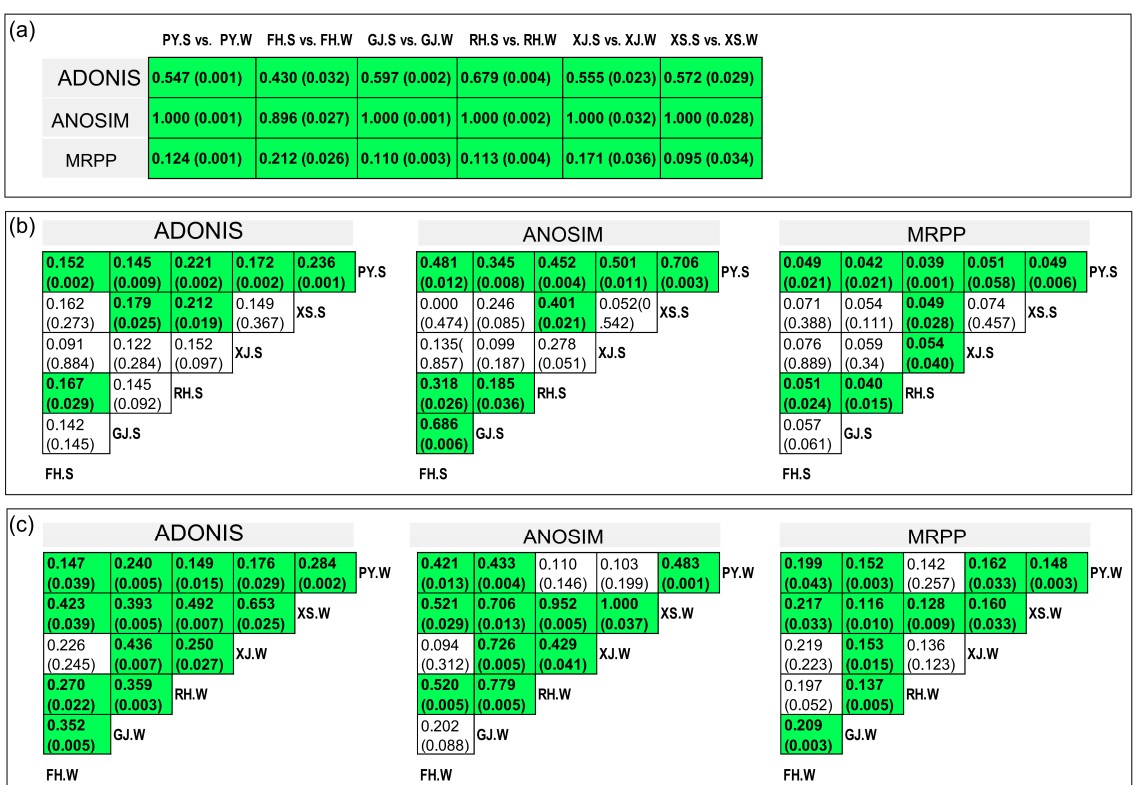

**Figure 4.** Pairwise dissimilarity tests of community composition (**a**) between sediment and water samples, (**b**) among sediments samples, and (**c**) among water samples. For ADONIS (analysis of variance using distance matrices), ANOSIM (analysis of similarity), and MRPP (multi-response permutation procedure analysis), the numbers outside the bracket are '$\delta$', 'R', and 'F', respectively. *p*-values are in bracket. Green cell represents significant difference ($p < 0.05$).

## 3.3. Drivers of Bacterial Community Variation

Redundancy analyses (RDA) evaluated the influences of environment nutrients on the bacterial communities in sediments (Figure 5a) and surface waters (Figure 5b). The results indicated that TN, $NO_3^-$, $NH_4^+$, and TP showed close relationships with the variation of bacterial communities in sediments (permutation test, $p < 0.05$, Figure 5a). The first two axes explained 43.56% of the taxonomic variances. The significant nutrient factors were TN, $NO_3^-$, TP, DOC, TN:TP, DOC:DIN (dissolved inorganic nitrogen), and DIN:SRP (permutation test, $p < 0.05$, Figure 5b) for the variation of bacterial communities in water. The first two axes explained 62.82% of the taxonomic variances. Nitrogen and phosphorus were important drivers in sediment bacterial communities. However, bacterial communities in surface waters were affected by nitrogen, phosphorus, and dissolved organic carbon, as well as their stoichiometric ratios, implying the influences of nutrient availability and resources quality on bacterial communities in surface waters.

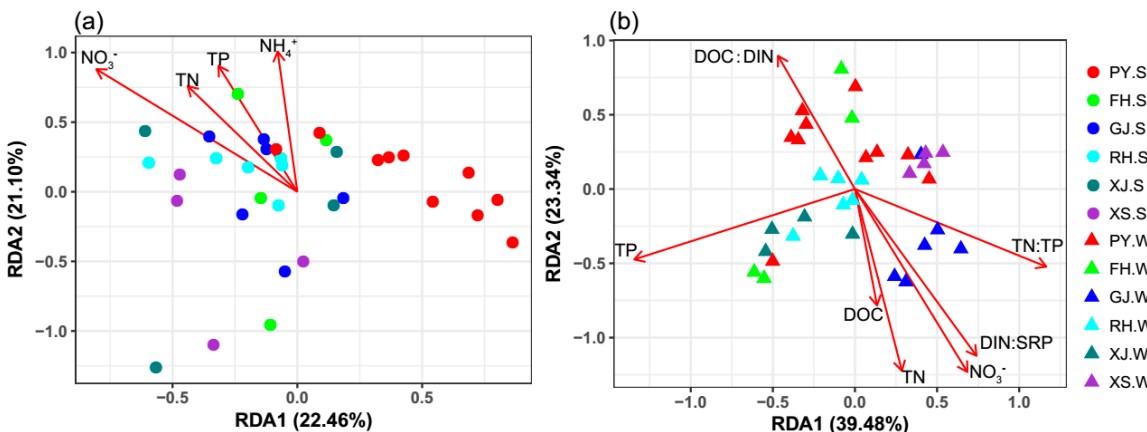

**Figure 5.** Biplot of multivariate redundancy analyses (RDA) showing the relationship between community composition and environment nutrients and nutrient ratios in (**a**) sediments and (**b**) surface waters. The red arrows denote the significant variables (permutation test, $p < 0.05$), including total nitrogen (TN), nitrate ($NO_3^-$), total phosphorus (TP), ammonium ($NH_4^+$) for sediment bacterial communities, and TN, $NO_3^-$, TP, dissolved organic carbon (DOC), TN:TP, DOC:DIN (dissolved inorganic nitrogen), and DIN:SRP (soluble reactive phosphorus) for bacterial communities in surface waters.

### 3.4. Co-Occurrence Network

Co-occurrence networks of bacteria in sediments and surface waters were built based on correlation relationships (Figure 6). Overall, 1129 nodes and 3993 edges composed the network of sediment bacteria, while 1043 nodes and 17,470 edges composed the network of water bacteria (Table 1). Topological parameters of the networks were calculated to describe the complex interrelationships between OTUs (Table 1). The distribution of node degree was well-fitted ($p < 0.001$) by the power law for both networks (Figure S2), indicating that the networks were scale-free and non-random. Network comparison that is based on average degree, average path length, and clustering coefficient showed that the bacterial network was more connected and complex in water than in sediments (Table 1). Moreover, both bacterial networks had "small world" properties and significant modular structures (Figure 6 and Table 1), because the modularity, network centralization, clustering coefficient, and average path length of the real networks were greater than those of their random networks. The bacterial networks in the sediments and surface waters were clearly parsed into seven and five major modules (with more than 50 nodes), respectively (Figure 6b and Figure S3). These modules had different taxonomic composition (Figure S3). Most of the modules had significantly higher niche width than the overall communities (Figure S4). According to the within-module connectivity (Zi) and between the module connectivity (Pi), there were 27 and five module hubs and two and three module connectors in the co-occurrence networks in sediments and surface waters, respectively (Figure S5 and Table S3).

To further investigate the influences of environmental variables on bacterial communities in sediments and surface waters, the correlations between environmental variables and individual OTUs were conducted (Figure S6). In the network of sediment bacteria, the environmental variables had 520 significant correlations (450 positive and 70 negative) that were associated with 418 OTUs (Figure S6). $NO_3^-$ had the highest number of correlations (257), most of which were positive, followed by TP, $NH_4^+$, OC:OP, TN, and others (Table S4). Module-A had the highest number of correlations, followed by -C, -E, -B, -D, -F, and -G. In the network of water bacteria, the environmental variables had 2212 significant correlations (1053 positive and 1159 negative), with 926 OTUs (Figure S6). TP had the highest number of correlations, followed by TN:TP, $NO_3^-$, DOC:DIN, DIN:SRP, and others (Table S5). TP had more negative correlations, whereas the N:P ratios (TN:TP and DIN:SRP) had more positive correlations. Module-A had the highest number of correlations, followed by -C, -E, -B, and -D. The results were consistent with RDA.

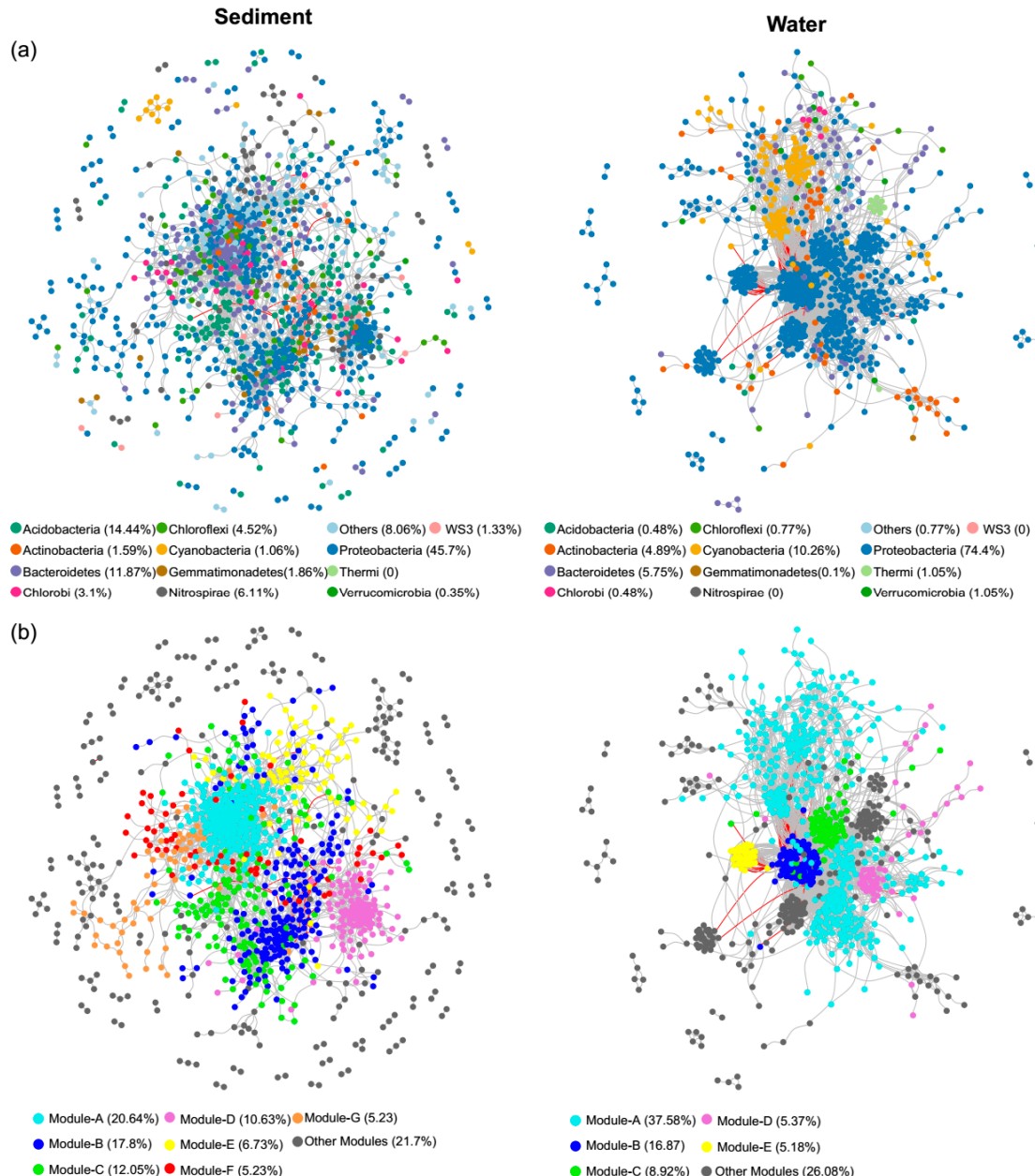

**Figure 6.** Co-occurrence networks. The nodes were colored according to (**a**) phyla (with relative abundance > 1%) and (**b**) major modules (with more than 50 nodes). Nodes represent the OTUs. Edges represent Spearman's correlation relationships. Only strong and significant correlations (Spearman's R > 0.8 or R < −0.8, $p < 0.05$) are shown. Positive and negative correlations are shown in grey and red lines, respectively.

**Table 1.** Topological parameters of the real co-occurrence networks and their associated random networks (permutation = 999, values shown mean ± SD). Comparisons between two real networks were conducted using *t*-test. Comparisons between real network and its corresponding random network were conducted using Z-test.

| Topological Parameters | Sediment | | Surface Water | |
| --- | --- | --- | --- | --- |
| | Real | Random | Real | Random |
| Number of Nodes | 1129 | 1129 | 1043 | 1043 |
| Number of Edges | 3993 | 3993 | 17,470 | 17,470 |
| Negative Edges | 20 (0.5%) | 20 (0.5%) | 109 (0.6%) | 109 (0.6%) |
| Average Degree | 7.073 | 7.073 | 33.499 | 33.499 |
| Average Path Length | 6.439 [a] | 3.810 ± 0.005 * | 4.997 [b] | 2.298 ± 0.001 * |
| Diameter | 21 [a] | 7.010 ± 0.175 * | 15 [b] | 3.000 ± 0.000 * |
| Clustering Coefficient | 0.372 [a] | 0.006 ± 0.001 * | 0.747 [b] | 0.032 ± 0.000 * |
| Centralization Degree | 0.052 [a] | 0.009 ± 0.001 * | 0.112 [b] | 0.019 ± 0.002 * |
| Centralization Betweenness | 0.073 [a] | 0.010 ± 0.002 * | 0.104 [b] | 0.002 ± 0.000 * |
| Modularity | 0.696 [a] | 0.347 ± 0.003 * | 0.727 [b] | 0.140 ± 0.002 * |

Note: * indicates the significant differences between the random network and the real network at the significant level $p < 0.05$ (Z-test). Different superscript letters (a and b) indicate significant differences between two real networks at the significant level $p < 0.05$ (*t*-test).

## 4. Discussion

In this study, bacterial communities in sediments had higher species richness and diversity than in water, as is consistent with global patterns of microbial diversity [4,50] (Figure 2). Much research has shown that ecosystems with more species are more efficient in removing nutrients from the environment than those ecosystems with fewer species [51–53], because more species can make the utmost of the niche opportunities, allowing for a greater proportion of bioavailable nutrients to be captured in metabolisms [52]. High bacterial diversity in sediments suggests sediments as the hotspot in nutrient metabolism and removal in the river-lake systems of Poyang Lake. However, because some species are functionally redundant in a community [51,54,55], it remains unclear how species richness and diversity influence nutrient cycling [52,56]. Thus, it is helpful to explore the functions of these bacterial communities in future studies in order to understand the mechanisms of nutrient biogeochemical cycling in Poyang Lake and its tributary rivers. For example, taxonomic composition and environmental variables determine the profiles of functional genes, resulting in variations of elemental cycling in estuaries and lake sediments [57,58]. Moreover, many factors have been known to cause heterogeneity in bacterial diversity in aquatic ecosystems, indicating the dynamic nature of the environment [59]. In our study, the alpha diversity in sediments was not significantly different between rivers and the lake, while, in the surface waters of rivers and the lake, the alpha diversity had some differences. The results suggested that bacterial communities in surface waters in the river-lake system of Poyang Lake are more dynamic than in the sediments.

In addition, our results showed that the bacterial communities were distinct between sediments and surface water, as well as between Poyang lake and its tributaries. The results suggested that the river-lake system of Poyang Lake provides a diverse and distinct habitat for bacterial communities, including lake water, lake sediment, river water, and river sediment. It is unequivocal that water and sediment are distinct habitat and harbor distinct bacterial communities. Moreover, lakes and rivers are typically subjected to different environmental conditions, such as flow velocity, water residence time, organic matter quantity and quality, and nutrients content, which can affect the compositions and functions of the microbial community [12,15,29,60]. In our previous study, we demonstrated that bacterial communities in stream biofilms were distinct from downstream lake water in both taxonomic and functional composition [29]. However, when comparing different tributaries, this study revealed that bacterial communities in surface waters were more different among tributaries.

As a eutrophic lake, it is important to know the interactions between nutrient dynamics and bacterial community variations in surface waters and sediments in the river-lake system of Poyang Lake. The results showed that nitrogen and phosphorus were important drivers for sediment bacterial

communities. However, nutrients and nutrient ratios affected the bacterial communities in surface waters. The results suggest that nutrient availabilities are important divers for bacterial communities in surface waters and sediments. Resource qualities are also important in shaping bacterial communities in surface waters. The availabilities and stoichiometric ratios of key chemical elements, such as C, N, and P, have been demonstrated to be essential in understanding bacterial diversity and community structure [2,61–64]. Microorganisms responded differently to nutrient concentrations and ratios, rooted in the ecological strategies and metabolic features of the responsive taxa [65]. As a result of excess fertilizer application [66] and sewage and septic inputs [67,68], agriculture and urbanization increase the nutrient concentrations and ratios of aquatic ecosystems, leading to reduced water quality [69] and altered aquatic communities [70,71]. In the Poyang Lake watershed, agriculture activities, urbanization, and industrialization enhance nutrient inputs that can be delivered by water and deposited in sediments, which strongly change the microbial communities across this river-lake system and lead to harmful algal blooms [72–74]. Most of the tributaries had higher nitrogen concentrations (Table S1) than the eutrophication threshold (TN = 0.65 mg/L) [75], and all of the tributaries had even higher phosphorus concentrations (Table S1) than the eutrophication threshold (TP = 0.03 mg/L) [75]. Thus, both the N and P inputs must be controlled to mitigate eutrophication in Poyang Lake.

Network analysis can provide profound and unique insights into highly complex microbial communities in this river-lake system, such as community assembly rules, potential taxon interactions, and shared physiologies [76]. The results showed that the bacterial networks in sediments and surface waters had non-random, scale-free, and "small world" properties, which are the structural characteristics of many microbial ecological networks [77,78]. The topological properties of the networks can offer more information of the network structure. For example, the average degree explains a complex pairwise connection, the average path length describes node distribution, the clustering coefficient describes the degree of nodes tend to cluster together, and the modularity index of the positive network indicates modular structures with a value larger than 0.4 [79]. The network of bacteria in water was more complex than the network of sediment bacteria. In general, communities with a lower diversity and high complexity of co-occurrence had lower stability [80,81]. Moreover, in respect of modularity, taxa played different roles in the co-occurrence network. For example, module hubs are highly linked nodes within a certain module, while connectors are linking nodes between different modules. In our study, a network of sediment bacteria had more module hubs and connectors than that of bacteria in surface waters. These module hubs and connectors (Table S3) play very important roles in maintaining network structures and community stability [82], and their disappearance leads to a breaking of modules and networks [83]. Thus, the bacterial communities in the surface waters were more vulnerable to various disturbances than in sediments due to lower alpha diversity, high complexity of network, and a small number of key taxa (module hubs and connectors).

In addition, a module is a group of highly interconnected nodes in a network with less linkages with nodes belonging to other modules [79,84]. Modularity reflects synergistic and competitive interactions, as well as niche differentiation and it is a characteristic of many complex systems [84,85]. Taxa can be grouped into modules due to functional complementarity, and the division of communities into modules provides insights into the responses of different groups of taxa in the communities [83]. Module-A, -C, and -E in the sediment bacterial network and Module-A, -C, -E, and -B in the water bacterial network were more strongly associated with some nutrient variables, while other modules had very few correlations with nutrients (Figure S6). Moreover, nutrients had more strong correlations with OTUs in network of bacterial communities in water. The results suggest that environment nutrients had stronger influences on bacterial communities in surface waters than in sediments, but were influenced differently on different groups of taxa in both water and sediment bacterial communities. Some of the modules are more susceptible to nutrient variations, contributing to the responses of the whole communities.

Our study examined the bacterial diversities and community structures, as well as the influence of nutrient factors on bacterial communities in different habitats of a river-lake system (Poyang

Lake). Many studies have demonstrated that more other environmental factors have the potential to influence bacterial communities in aquatic ecosystems, such as pH, salinity, temperature, hydrology, geomorphology, and land cover, etc. These environmental variables should be considered in future study. Moreover, other pollutions should also be concerned, such as heavy metals and polycyclic aromatic hydrocarbons, which can also affect bacterial communities in aquatic environments. In addition, Poyang Lake has a profound water level fluctuation around the year. In this study, we only undertook this intensive investigation in August, which can represent the status of Poyang Lake at high water level. Thus, more intensive intra-annual sampling and more environmental variables would be useful in establishing a comprehensive understanding of bacterial communities in Poyang Lake.

## 5. Conclusions

Investigating bacterial communities that live in surface waters and sediments of the river-lake system of Poyang Lake is pivotal in understanding the structures and the diversity of this freshwater ecosystem. This study highlights that the river-lake system of Poyang Lake provides diverse habitats, including river sediment, river water, lake sediment, and lake water, which harbor distinct bacterial assemblages. Nutrients were closely associated to bacterial community variations in sediments, while nutrients and nutrient ratios were closely associated to the bacterial community variations in surface waters. Moreover, bacterial communities in the surface waters had lower alpha diversities but a complex co-occurrence network than in sediments, and are thus more vulnerable to environmental disturbances. This study suggests that both nutrient availability and resources quality are important factors that drive bacterial communities in this highly linked river-lake system, especially in surface waters. Further research is required to know the potential influences of nutrient variations on biogeochemical processes in this system by identifying bacterial community functions.

**Supplementary Materials:** The following are available online at http://www.mdpi.com/2073-4441/11/5/930/s1, Figure S1: Relative abundance of microorganisms at family level in lake sediment (LS), river sediment (RS), lake water (LW), and river water (RW) in the river-lake system of Poyang Lake. Only the families that had a relative abundance > 1% in either habitat are shown, Figure S2: The node degree distributions of real co-occurrence network (colored) and random networks (grey) of bacterial communities in sediments and surface waters, Figure S3: Composition of modules in (a) sediments and (b) surface waters. Pie charts showing the proportion of nodes number in taxonomic groups, Figure S4: Niche width value of bacteria in (a) sediments and surface waters, as well as different modules in the co-occurrence network of (b) bacterial communities in sediments and (c) bacterial communities in surface waters. The top, middle, and bottom lines of the box indicate the 75th quartile, median, and 25th quartile, respectively. The whiskers above and below the box indicate the minimum and maximum. The black dots indicate outliers. Different letters above each box indicate significant differences (ANOVA, $p < 0.05$), Figure S5: Zi-Pi plot indicates the topological roles of OTUs in the network. Each dot represents an OTU colored by taxonomic groups. The topological roles were determined according to their connectivity, within-module connectivity ($Zi = 2.5$) and among-module connectivity ($Pi = 0.62$). The detail of the module hubs and the connectors are shown in Table S3, Figure S6: Connectedness between nutrient variables and operational taxonomic units (OTUs) in the co-occurrence network of bacterial communities in sediments and surface waters. The nodes were colored according to (a) phyla (with relative abundance > 1%) and (b) major modules according to Figure 6. Black nodes represent nutrient variables. Other colored nodes represent OTUs. Edges represent Spearman's correlation relationships (Spearman's $p < 0.05$). Positive and negative correlations are shown in grey and red lines, respectively, Table S1: Nutrients contents (mg/g) and stoichiometric ratios (molar ratio) of lake water and river water in the river-lake system of Poyang Lake. The values represent Mean ± SD (standard deviation), Table S2: Nutrients contents (mg/g) and stoichiometric ratios (molar ratio) of lake sediment and river sediment in the river-lake system of Poyang Lake. The values represent Mean ± SD (standard deviation), Table S3: List of module hubs and connectors in co-occurrence networks according to the connectivity of each node, Table S4: Number of significant correlations (Spearman's $p < 0.05$) between environmental variables and OTUs in the network of bacteria in sediments. 'Pos' represents positive correlation. 'Neg' represents negative correlation, Table S5: Number of significant correlations (Spearman's $p < 0.05$) between environmental variables and OTUs in the network of bacteria in surface waters. 'Pos' represents positive correlation. 'Neg' represents negative correlation.

**Author Contributions:** Z.R. and X.Q. designed the study, did the analyses, and prepared the manuscript; X.Q. and Y.Y., performed the field work and laboratory work; M.Z. and W.P. gave suggestions during the whole work.

**Funding:** This study was funded by the National Key Project R and D Program of China, grant number 2017YFC0404506; the National Natural Science Foundation of China, grant number 51439007, 41671048, and 51779275; the Project of State Key Laboratory of Simulation and Regulation of Water Cycle in River Basin, grant number SKL2018CG02; and the IWHR Research and Development Support Program, grant number

WE0145B532017. The APC was funded by the National Key Project R and D Program of China, grant number 2017YFC0404506.

**Acknowledgments:** We are grateful to the anonymous reviewers for the comments, to Yuhang Zhang and Chenyu Yang for their assistances in the field and laboratory work.

**Conflicts of Interest:** The authors declare no conflict of interest.

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
