# Peer review of "Nutrients Drive the Structures of Bacterial Communities in Sediments and Surface Waters in the River-Lake System of Poyang Lake"

_water, doi:10.3390/w11050930_

Round 1

Reviewer 1 Report

see attached file

Reviewer 2 Report

General overview

Authors of this manuscript have presented a study related to bacterial communities in sediments and water in the river-lake system of Poyang Lake, the largest freshwater lake in China. Bacterial community structure, influence of nutrients on bacterial communities’ composition was investigated and co-occurrence networks of bacteria in sediments and water were built. The topic is interesting and an important issue.

However, there are a lot of spelling and grammar errors through the manuscript, proofreading is definitely needed to improve the English.

Also there is a large number of formatting issues and an expansion of the methods is required before it is acceptable for publication.

Definitely, after the authors make respective correction and improvement, the manuscript deserves to be published in the Water journal.

Specific comments:

References № 6, 7, 14, 15, 59 are lacking in the article body, please delete these articles from the section “References”.

References № 32, 33 and 34 follow the reference № 70 in the article body, not № 31. Please correct.

Line 120-121. Method EPA 300.0 doesn’t cover the determination of total nitrogen. Please give correct reference on the method with revision number and year (for example: EPA 300.0, revision 2.1, 1993). Also please give references on the other methods used for chemical analysis with revision number and year.

Line 125. Give the abbreviation expansion for the term “DOC” as you did for TN, TP, RSP etc.

Line 162. Phylogenetic distance is not an alpha diversity index, please explain what did you mean.

Figure 4 can be placed into the supplementary materials.

Line 232. Give the abbreviation expansion for the term “DIN”.

Figure 5 (a). Write NH4 instead of HN4.

Line 247-248; 342-343. Positive correlation doesn’t necessary indicate on the cooperation relationship between OTUs. It can be something else that affects both OTUs. That’s why you should write in this way: “It can indicate on…” or “There is a chance of existance of a correlation realationship…”.

Figure S1. Please give an explanation what are (a), (b) and (c) in the figure caption.

Figure S3. Write OTU instead of OUT.

Reviewer 3 Report

Dear Xiaodong Qu,

the results reported in this manuscript are interesting and new but  further efforts have to be made. I suggest to add more information on the studied environment adding a result section where in brief Authors describe changes in the nutrient concentration in the different environments analysed, also comparing in the discussion the trophic conditions with data from the literature. The discussion has to be re-organised emphasising the most notable results, now explained in the last part of the discussion, and cutting the well-known information in the first part. The Discussion has to be enriched with comments trying to link the presence of some phyla to specific environmental conditions, also reporting other Authors’ data. The comparison with other similar results from the literature would give to this work a wider significance beyhond regional interest .

Comments:

TITLE : nutrient driving bacterial community diversity (or structure) (?)…..

INTRODUCTION:

-which is the authors’ hypothesis?

ROW 76-78: This information is already reported in the Material and methods section

ROW 234: do you mean nitrogen as NO3 ? In the fig. 5a I see that also TP seems to be important.

Fig.5: please in the fig 5b change HN4 in NH4. Please check also tab s4.

Fig. s4: please avoid similar colors to identify clusters in the figures. Change colors for acidobactiria  and proteobacteria ; cloroflexi e verrucomicrobia. They make these groups undistinguishable. Check also for Fig.5

M&M

Row 151: Analyses …of what? Please complete.

RESULTS

The fig. 4 also show that changes in nutrients, including TP and DIN:SRP ratio, can change the community structure. Why only Nitrogen? Then is there some limitation?

The tab. S1 and tab S2 report differences in the nutrient concentrations that are a key parameter to explain the microbial structure. Why don’t you describe briefly these environments on the base of nutrient concentrations?

235-236 please explain better this sentence. The nutrient ratio change implies for bacteria changes in the quality of the resources and in the nutrient availability.

DISCUSSION

In this section there is a lack of comments on the relationships occurring between major groups found and environmental conditions (different nutrient availability). Is it possible to explain some relationship between diversity and different nutrient availability (i.e. DOCDOC:DIN; TN:P; DIN:SRP etc.)? The observed different distribution of phyla can let infer different ability in the resource exploitation. Are there information from the literature?

Row 288-291 : Please check this sentence. High microbial diversity is sustained by a wide variety of food sources. Low diversity occur in those environments with scarce variety of food.

Row 291 Poyang lake is the largest …..already told inmany way in th eanuscript , also in the title

Rows 300 and 302 : “water column”   I suggest “surface water”

Rows 307 and 308 this is well known. Please avoid these two sentences

Row 316: “suggesting that sediments in this river-lake”…I think that this is already known

Rows 318-328: these observations are mostly obvious and repeated through the manuscript. Please try to describe the novelty of this survey

Row 362- 364: please change this sentence leaving only what is new from this survey.

Reviewer 4 Report

In general, this manuscript is well-written. However, I have some major and minor comments.

Major comments:

It is well-known that (heavy) metals and PAHs can affect bacterial communities in lakes. Consequently, I recommend that future studies should not only analyze nutrients and their effects on bacterial communities.In addition, only one time point was analyzed. This limits the significance of the results and should be mentioned in a study limitation section or in the discussion.

I was not able to find the final OTU table used for all analyses. These data should be included as supplementary table. The environmental data should be included in the manuscript and not as Table S2. In addition, the authors should analyze differences of the environmental data among the sites.

Some parts of the discussion belongs to the introduction (first paragraph). Other parts are more like a review (lines 307-312). The authors should discuss their data with recent research and what these results mean for the management.

Minor comments:

Lines 16/ 18 and others: "the" can be deleted here, but should be added in other lines, e.g., line 28: " in the water column"

Line 23: more diverse taxa or bacterial communities in general?

Line 26: driver

Line 27: which nutrients?

Line 34: increased our current knowledge

Line 41: environments

Line 46: delete "some"

Line 50: not sure what is meant by "different chemical and physical environment - water and sediment? or aquatic ecosystems?

Line 109: water and sediment samples.... (the first "samples" can be deleted)

Line 107: As far as I understood, there is only one sample per site (and not a composite sample of three subsamples?)

Line 121 and others: I miss information on a webpage (there are different methods) and which analyzer/ company was used.

Line 134: How many replicates/subsamples were measured per site?

Line 140: of the 16S rRNA gene

Line 148: which run/kit?

Line 152: I was not able to find the raw data. I guess the authors meant SRP133903.

Line 160: which tools were used for the calculation of the alpha diversity indices? How was the tree calculated used for phylogenetic distance? The reference for Greengenes is missing. In addition, there is no information on how non-bacterial sequences were removed.

Line 165: How was tested if all assumptions for an ANOVA were met? The sample number is not equal for the lake and its tributaries. As consequence, I recommend that the authors use a LMER instead an ANOVA. In addition, there is no information which R packages and commands were used for the statistical analyses (e.g., RDA, MRPP).

Line 180: Why a t-test and not a test for multiple comparison with p value adjustment?

Line 184: I miss information on the coverage and how many sequences were lost in each processing step. 

It is well-known that sediment bacterial communities are more diverse than water communities. Thus I recommend that the authors also perform separate analyses for sediment and water communities to identify differences within the different habitats (and not only between them).

Line 197: The dominant bacterial phlya in LS and RS were.... might be better than the current sentence

Line 209: in both habitats

Line 213: this belongs to the discussion

Line 216: An analysis on phylum level is not very interesting. I recommend that the authors repeat their analysis using family or order level.

Line 226: envFit is not a statistical analysis. It might be better to add a Permanova or another statistical test here.

Other important drivers are water temperature and salinity. I am not sure why these factors were not determined in this study.

Line 235/371: driver

Line 285: it might be better to show only the statistically significant drivers. 

Line 285: Consistent with (?)

Line 294-302: which is known for other lakes and stressed environments (e.g., estuaries or coasts). Some recent studies should be added and discussed here.

L. 295: there are many tools available to assess functional changes based on 16S rRNA gene data (e.g., PICRUSt, Tax4Fun2,... as performed in their previous study). I recommend that the authors include this analysis.

Line 349: which might be related to the higher diversity?

Line 364: which means what?

Conclusion: The conclusion is more a summary. I miss information on how others can use the data for management strategies and a short outlook for future studies.

Round 2

Reviewer 3 Report

Dear Authors,
I think the manuscript is improved. There is still an issue with the title. I think that the new title you propose is not correct from a point of view of grammar. It may be better: Nutrients drive the structure of the bacterial communities in sediments and surface waters in the river-lake system of Poyang Lake. Please chek it.

Author Response

Dear Reviewer,

Thanks for your suggestion. We revised the title to "Nutrients drive the structures of bacterial communities in sediments and surface waters in the river-lake system of Poyang Lake". We think the "THE" before "bacterial communities" is not needed. 

Reviewer 4 Report

I thank the authors for the revised version of the manuscript. I have only some minor comments:

Figure 5: I meant that only the statistically significant factors (coloured in red) are shown in the figure. In addition, one of the last 2 sentences can be deleted because it is redundant. 

Conclusion, line 384: functions were not analysed in this study. I recommend that the authors change this sentence to "is pivotal to understand the structures and the diversity..."

UCHIME version?

Author Response

Dear Reviewer,

Thanks for your comments. We addressed the issues as below:

Figure 5: I meant that only the statistically significant factors (coloured in red) are shown in the figure. In addition, one of the last 2 sentences can be deleted because it is redundant.

Response: The last sentence was deleted. The figure was revised with only significant factors.

Conclusion, line 384: functions were not analysed in this study. I recommend that the authors change this sentence to "is pivotal to understand the structures and the diversity..."

Response: Revised.

UCHIME version?

Response: Revised with “version 4.2”